# Digital Soil Mapping of Cadmium: Identifying Arable Land for Producing Winter Wheat with Low Concentrations of Cadmium

Karl Adler [1,*], Kristin Persson [1], Mats Söderström [1], Jan Eriksson [2] and Carl-Göran Pettersson [3]

1   Department of Soil and Environment, Swedish University of Agricultural Sciences, 532 23 Skara, Sweden
2   Department of Soil and Environment, Swedish University of Agricultural Sciences, 750 07 Uppsala, Sweden
3   Lantmännen R&D, 104 25 Stockholm, Sweden
*   Correspondence: karl.adler@slu.se

**Abstract:** Intake of cadmium (Cd) via vegetable food poses a possible health risk. Cereals are one of the major sources of Cd, and the Cd concentration in the soil has a great effect on the levels in the grain. The aim of the study was to produce decision support for identification of areas suitable for low-Cd winter wheat production in the form of a detailed digital soil map covering an important agricultural region in southern Sweden. A two-step approach was used: (1) we increased the number of soil Cd observations by combining two sets of soil samples, one with laboratory Cd analyses (304 samples) and one with predicted Cd from a portable x-ray fluorescent (PXRF) sensor (2097 samples); and (2) a digital soil mapping (DSM) model (gradient boosting regression) was calibrated on all 2401 soil samples to create a soil Cd concentration map using a number of covariates, of which airborne gamma ray data was identified as the most important. In the first step, cross-validation of the PXRF model obtained a model efficiency (E) of 0.82 and mean absolute error (MAE) of 0.08 mg kg$^{-1}$. The DSM model had an E of 0.69 and MAE of 0.11 mg kg$^{-1}$. The map of predicted soil Cd concentrations were compared against 307 winter wheat (*Triticum aestivum* L.) grain samples with laboratory-analyzed Cd concentrations. Areas in the map with low soil Cd concentrations had a high frequency of lower grain Cd concentrations. The map thus seemed to have potential for finding areas suitable for production of low-Cd winter wheat; e.g., for baby food.

**Keywords:** cadmium; digital soil mapping; machine learning; winter wheat; PXRF





## 1. Introduction

Cadmium (Cd), a heavy metal that is present in varying concentrations in agricultural soils, has been of interest in medical research for more than 70 years [1,2]. It is regarded as a toxic substance that is detrimental to human health, so intake should not exceed 2.5 μg kg$^{-1}$ per week according to European legislation [1]. For example, links have been shown between the ingestion of Cd and bone malformation and brittleness, kidney damage, and increased cancer risk [3–5]. A connection between Cd exposure and cardiovascular disease has also been established [6]. Food crops; e.g., wheat (*T. aestivum* L.), can accumulate Cd in their roots, shoots, and most importantly, grain [7,8]. A main factor that determines Cd uptake is the soil Cd concentration [9,10]. Hence, there is often a direct link between the Cd content in agricultural soil and the food chain, although other variables such as the soil pH can influence plant uptake significantly [11]. The high consumption of grain products makes grain one of the major pathways for Cd exposure in humans [1,7,12].

Under European Union regulations, the maximum permissible concentration of Cd in most types of cereals is 100 μg kg$^{-1}$ wet weight, including winter wheat [13]. For cereal-based baby food, the limit is 40 μg kg$^{-1}$. Regarding agricultural soils, there are few legal limits or guidelines for allowed concentrations of Cd.

There are no readily available maps with sufficiently high resolution that can be used as decision support in identifying favorable areas in Sweden for specific food production

forms. However, some studies have investigated Cd concentrations on national, regional and local (within-field) scales [14,15]. The results showed that Cd concentrations in Swedish soils are highly variable at both regional and field scales. In Sweden, concentrations of Cd in soil are often correlated with the influence of sedimentary bedrock such as alum shale or zinc-rich Cambrian sandstone. Areas dominated by heavy clay soils also tend to have higher concentrations than average [9]. Anthropogenic input such as atmospheric deposition and application of phosphorus fertilizer have also affected soil Cd concentrations [9]. In the county of Skåne, a major grain-producing area in Sweden, the Cd concentrations in soil and in winter wheat grain are generally higher than the national average. Maps of Cd concentrations in agricultural soils in Skåne could be a useful tool for farmers, the grain industry, and policymakers seeking to identify suitable areas for winter wheat production from a Cd perspective. In particular, areas suitable for production of winter wheat intended for baby food can be challenging to find.

In order to create useful maps of soil Cd concentrations, a large and spatially extensive calibration dataset of soil samples with laboratory-determined Cd concentrations is needed, but this can be expensive and time-consuming to achieve. Portable X-ray fluorescence sensing (PXRF) used ex situ can be a fast and sufficiently accurate alternative method to conventional laboratory analysis, especially when basic sample-preparation steps have been taken such as drying, sieving, and grinding [16–20]. By applying machine learning methods (random forest, multivariate adaptive regression splines (MARS), and multiple linear regression), elements easily quantified with a PXRF device can be used for predicting concentrations of other elements such as Cd that are difficult to measure accurately due to the relatively low concentrations encountered in soils [16]. The speed and accuracy of PXRF measurements makes the method favorable for mapping purposes [21,22]. Maps of soil properties can be created in digital format using digital geographical information in an approach known as digital soil mapping (DSM) [23]. These maps can be created using geostatistics such as regression kriging or machine learning models such as regression trees [24]. The use of machine learning algorithms for DSM has become increasingly popular in recent years [25].

The overall aim of the present study was to outline and test a method for mapping soil Cd concentrations to provide decision support for the grain industry or farmers to target soils suitable for production for winter wheat with low Cd concentrations; e.g., for the manufacturing of baby food.

The specific objectives of the study were to:

1. Increase the size of the calibration dataset for the DSM model using predictions from PXRF measurements and test whether the new larger data set was better than solely using data from wet chemistry analysis for DSM model calibration.
2. Employ a DSM model to create a detailed map of soil Cd (with a 50 m spatial resolution and 90% prediction intervals) using a machine learning algorithm with various covariates and covariate importance metrics and then evaluate the model's performance by comparing its results with lab-analyzed Cd concentrations.
3. Assess the applicability of the soil Cd map in identifying areas suitable for low-Cd winter wheat production by comparing winter wheat grain Cd concentrations in different parts of the map.

## 2. Materials and Methods

### 2.1. Study Area

The study area was the county of Skåne, which is in the southernmost part of Sweden and covers approximately 11,035 km$^2$ (Figure 1). Skåne is one of the leading agricultural regions in Sweden, with 45% of the land area used for agriculture. It is one of the dominant production areas for cereal crops such as wheat and barley (*Hordeum vulgare* L.) but also for sugar beet (*Beta vulgaris* L.), potatoes (*Solanum tuberosum* L.), oilseed rape (*Brassica napus* L.), and forage crops, plus many others [26].

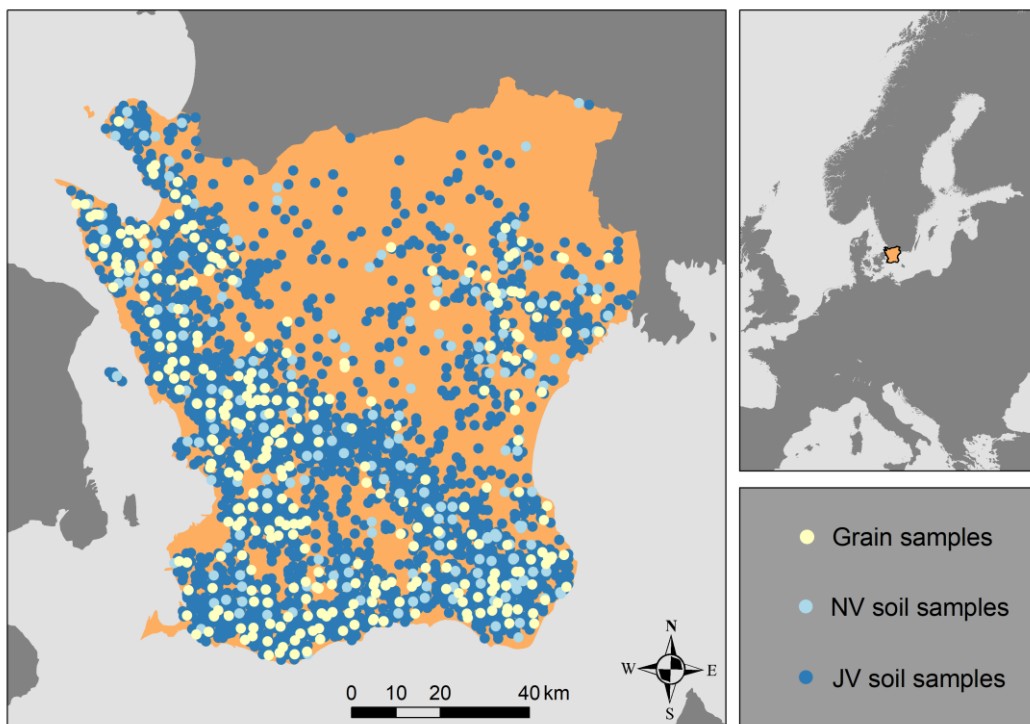

**Figure 1.** Maps showing the location of the study area (Skåne County in southern Sweden) and the data-collection points. NV (Naturvårdsverket) soil-sampling sites (*n* = 304), from which laboratory-measured cadmium data were used to calibrate the portable X-ray fluorescence and the digital soil mapping (DSM) model, are presented as light blue dots and JV (Jordbruksverket) sampling sites (*n* = 2097) as dark blue dots. Combined data from the JV and NV samples were used to calibrate the DSM model. Sites from which laboratory data on cadmium concentrations in wheat grain samples were available (*n* = 307) are presented as light yellow dots.

Skåne has different forms of bedrock linked with the Törnquist Zone, a fault zone that runs diagonally across the county from northwest to southeast and principally separates Precambrian granite and gneiss in the north and northeast from Palaeozoic and Mesozoic sedimentary bedrock in the south and southwest [27,28]. Tertiary limestone dominates the southwest, and Cretaceous bedrock occurs in parts of northeast Skåne. The southeast is dominated by Cretaceous and Cambro-Silurian sedimentary bedrock forms such as sandstone, shales (including alum shale), and limestone. Jurassic shales dominate in the northwest with minor incursion of Triassic sandstone. The surface layers of loose deposits were formed during and after the Weichselian glaciation. Till clays dominate in the south and southwest, coarser-textured sandy tills in the central and northern parts, and postglacial sandy soil in the east, while postglacial clay is found in the northwest. Varying subareas of glaciofluvial sediments are commonly found, particularly in the center and in parts of northern Skåne [29].

Skåne is a highly productive agricultural region but has higher average soil Cd concentrations than other parts of Sweden. The higher Cd concentrations in Skåne can mostly be attributed to Cd-rich soil parent material [15]. Eriksson [14] showed that Cd concentrations in winter wheat grain from Skåne tend to be higher than the national average. Descriptive statistics on topsoil (0–20 cm depth) properties in agricultural land in Skåne calculated from the database of the Swedish national soil monitoring program for agricultural soil [14] are presented in Table 1.

**Table 1.** Descriptive statistics on soil pH; soil organic carbon content (SOC); cation exchange capacity (CEC); and clay, silt, and sand content in 332 topsoil samples (0–20 cm depth) from agricultural soils in Skåne.

| Property | Min | Max | Mean |
|---|---|---|---|
| pH ($H_2O$) | 4.9 | 8.0 | 6.5 |
| SOC (%) | 0.7 | 50 | 2.7 |
| CEC ($cmol_c\ kg^{-1}$) | 6.4 | 156 | 17 |
| Clay (%) | 1 | 55 | 14 |
| Silt (%) | 4 | 77 | 31 |
| Sand (%) | 2 | 94 | 55 |

*2.2. Soil Samples and Cd Analyses*

Two sets of topsoil samples (0–20 cm depth) that were collected in projects funded by the Swedish Environmental Protection Agency (Naturvårdsverket, NV) and the Swedish Board of Agriculture (Jordbruksverket, JV), respectively, were used in this study. The NV set consisted of 282 soil samples from Skåne that were mainly collected in 2011–2017 in a monitoring program of Swedish agricultural soil [14] plus 22 soil samples collected by Söderström and Eriksson [15] that were added to the NV dataset. Laboratory analysis data were available on these soil samples. These data comprised Cd pseudototals measured after extraction with 7 M $HNO_3$ in an autoclave at 120 °C for 30 min and determined via inductively coupled plasma mass spectrometry (ICP-MS) in accordance with Swedish standard SS 28311 [30]. The NV dataset was used as the calibration dataset for the PXRF and DSM models.

The purpose of the JV project was to provide access to a larger set of soil samples that were complementary to those in the NV project mainly in order to enable more detailed mapping of the soil phosphorus content. The NV and JV soil samples were taken in a predetermined grid of 1 km with a small random displacement of soil sample locations [31]. The JV dataset consisted of 2097 soil samples from Skåne. Cadmium was not analyzed for the JV soil samples. The JV dataset was used to expand the calibration dataset with PXRF-determined Cd-content values for the DSM model, but was never used for validation. Both the NV and JV sets were subsetted to only contain soil samples with a soil organic matter content below 20%; i.e., non-organic soils.

*2.3. PXRF Methodology*

A Niton XL3t GOLDD+ PXRF device (Thermo Scientific, Billerica, MA, USA) was used to analyze all soil samples ex situ. The PXRF device had a geometrically optimized large area drift detector with a silver (Ag) anode specified at 50 kV and 200 µA. The soil samples in the NV and JV datasets had already been dried, ground, and sieved (<2 mm) in accordance with recommendations for ex situ analysis with a PXRF device [32,33]. The measurement time was set to 180 s. Each soil sample was placed in a 32 mm double-ended XRF sample cup with a 4 µm-thick transparent polypropylene XRF film and set on the PXRF aperture. The measured elements used to predict Cd in the JV samples were those used previously in Adler et al. (2020); i.e., lead (Pb), cesium (Cs), zinc (Zn), vanadium (V), rubidium (Rb), strontium (Sr), zirconium (Zr), barium (Ba), manganese (Mn), titanium (Ti), calcium (Ca), iron (Fe), and potassium (K). These 13 elements were chosen because they occur in concentrations clearly above the limit of detection for the PXRF device in a vast majority of soil samples [16]. Recovery rates for the elements are presented in Adler et al. [16]. The recovery rates showed that the PXRF measurements were close to the reference sample (National Institute of Standards and Technology (NIST) 2709a) apart from Cs and Pb. However, the standard deviation in the recovery rates showed that the measurements were stable over time and thus suitable for modeling [16].

### 2.4. Grain Samples

Two datasets of Cd concentrations in winter wheat grain (*n* = 307) were used to investigate whether wheat grain with low Cd concentrations was mainly produced in soils with predicted low Cd concentrations. The grain samples for the first dataset were collected in 1992 (*n* = 196) and those for the second dataset in 2001–2017 (*n* = 111). The second set included data from both the first and second campaigns in the Swedish monitoring program. Thus, for part of the locations, grain data from the first sampling campaign (2001–2007) were matched to soil data from the second sampling campaign (2011–2017). In both cases, the grain samples were digested using concentrated $HNO_3$. For all samples except the ones collected in 2001 and 2003, the digestion was performed in a microwave oven at 120 °C for 60 min. For the samples collected in 2001 and 2003, the digestion was performed using a Tecator block heated to 135 °C for four hours and thereafter to 100 °C for 60 min. A comparative test showed that the measured Cd concentrations were similar for the two digestion methods [34]. The Cd concentration in the samples from 1992 was determined via inductively coupled plasma with mass spectrometry (ICP-MS), while the Cd concentration in samples from 2001–2017 was determined via inductively coupled plasma with sector field mass spectrometry (ICP-SFMS). Both datasets of grain samples should be representative because Cd concentrations in Swedish agricultural soil and winter wheat grain have been stable over the last two decades [14].

### 2.5. DSM Covariates

A range of covariates were used in the DSM model. A total of four different types of data were used to derive covariates: (i) airborne gamma-ray spectrometry data, (ii) a digital elevation model (DEM), (iii) quaternary deposit maps, and (iv) biogeochemical data (Table 2). Some covariates were present in a 50 m resolution grid that was previously constructed by Piikki and Söderström [35] that covered all agricultural land in the study area. Three new covariates were added to this point grid in the present study.

**Table 2.** Covariates from airborne gamma ray data, topographical data (digital elevation model, DEM), quaternary deposit maps, and biogeochemical data used in the digital soil mapping (DSM) model. Data on covariates marked with an asterisk (*) were prepared for the study by Piikki and Söderström [35]. Each soil class was converted to a separate covariate using one-hot encoding.

| Covariate | Type |
|---|---|
| Thorium (Th) * | Gamma-ray remote sensing |
| Potassium (K) * | Gamma-ray remote sensing |
| Uranium (U) * | Gamma-ray remote sensing |
| Topographic wetness index (TWI) | DEM derivative |
| Convergence index (ConvInd) | DEM derivative |
| Topographic position index (TPI5) (5 ha) * | DEM derivative |
| Topographic position index (TPI50) (50 ha) * | DEM derivative |
| Topographic position index (TPI500) (500 ha) * | DEM derivative |
| Elevation * | DEM |
| BioGeo | Cokriged biogeochemical data |
| Soil texture class: Clay, silt, clay till, till, sand, and other * | Quaternary deposit maps |

The airborne gamma-ray spectrometry data, which were provided by the Geological Survey of Sweden (SGU), comprised measured activity concentrations of thorium-232 (Th), potassiun-40 (K), and uranium-238 (U). Söderström and Eriksson [15] showed that these types of data, especially measured U concentrations, correlate well with Cd concentrations in agricultural soil in southeast Skåne. A full description of the data preprocessing can be found in the study by Piikki and Söderström [35].

A 2 m resolution DEM in raster format that was provided by Lantmäteriet (Swedish National Survey, Gävle, Sweden) was resampled to a 10 m resolution. The covariate grid created by Piikki and Söderström [35] contained the elevation and topographic position

index (TPI), which is the relative elevation of a raster cell compared with its neighbors within a specific area, in this case circular surrounding areas covering 5 ha, 50 ha, and 500 ha. Two more DEM derivatives were prepared for this study—the topographic wetness index (TWI) and the convergence index (ConvInd)—to provide a variety of DEM derivatives for mapping because the Cd concentration has been shown to be related to terrain gradients [36]. In short, TWI aims to describe the potential of a raster cell to accumulate water as a function of the flow accumulation and slope [37], while ConvInd uses a moving kernel window of a specific size (3 × 3 cells) to compute whether the area is concave or convex as a function of the slope and aspect [38].

The biogeochemical data provided by SGU comprised Cd concentrations in roots from various sedges (*Carex* L.), whole plants of water moss (*Fontinalis* Hedw.), and roots from meadowsweet (*Filipendula ulmaria* L.) sampled from small streams (*n* = 290) [39]. Cokriging was done using the Zn concentration from the same dataset as the covariate (*n* = 1596) as exemplified in Skåne by Rosenbaum and Söderström [40]. This was done in order to obtain a rough spatial pattern of environmental Cd concentrations in the study area. The samples used to obtain the biogeochemical data were uniformly distributed across Skåne.

Lastly, a legacy soil map on quaternary deposits with six texture classes was used as a covariate (Table 2). This soil map represented the general spatial distribution of soil classes at a depth of 50 cm. A more detailed description of the soil map was given by Piikki and Söderström [35].

### 2.6. Software

Calibration, validation, prediction of Cd concentration models, and other analyses were conducted in the Python programming language with the machine learning package Scikit-learn (version 0.22.2) [41] and the Py-earth package (version 0.1.0). Conversions between feature data and raster data and creation of an overview map were conducted in ArcGIS (version 10.6) (ESRI, Redlands, CA, USA). Cokriging was conducted using the Geostatistical analyst package in ArcGIS using simple cokriging with the default settings. The covariates TWI and ConvInd were computed using the grass GIS extension in QGIS (version 3.14) (QGIS Association, Switzerland). Plotting was conducted in Python using the packages matplotlib (version 3.1.3) [42] and datashader (version 0.12.1).

### 2.7. PXRF Model

In a study by Adler et al. [16], predictions of Cd concentration from PXRF measurements using random forest (RF) and multivariate adaptive regression splines (MARS) proved to be most accurate when compared against laboratory-analyzed concentrations. In the present study, the predicted output of RF and MARS was averaged to obtain an ensemble prediction of Cd concentration from the PXRF measurements. This was done to obtain an influence of both models in the prediction because MARS and RF are different models. For example, MARS can extrapolate while RF cannot [43]. Both models were set at the default settings because the initial testing showed no notable accuracy improvement with hyperparameter optimization (using a grid search). The models were solely calibrated and validated on the NV dataset, and the resulting model was applied on the JV dataset in order to obtain Cd concentrations for the JV dataset. Negative predicted concentrations were set to the lowest positive concentration predicted because the MARS model was able to extrapolate to negative concentrations.

### 2.8. DSM Model

Gradient boosting regression (GBR) was chosen to create the DSM model because it is (i) non-linear, (ii) a tree-based model that has an inbuilt feature importance metric, and (iii) applicable to loss functions and thus the creation of prediction intervals using quantile loss. The idea behind GBR is to create an ensemble of shallow trees (stumps) in a sequence with each tree fitted to minimize a loss function [44]. GBR with least squares loss was used as the actual DSM model, and GBR was used for the lower and upper quantile

using quantile loss. The 5th and 95th quantiles were chosen for the quantile loss functions. This framework enabled actual prediction (using least squares loss) and the 90% prediction interval (using quantile loss).

Selected hyperparameters of the DSM model were optimized using a grid search because the initial testing revealed an improved accuracy with tuning. The hyperparameters were learning rate (shrinkage factor), max depth (the maximum depth allowed in each regression tree), max features (the fraction of features considered at each split), minimum samples (the minimum number of samples needed for the creation of a leaf node in each regression tree), and subsampling (the fraction of samples used to construct each tree) (Table 3). Stochastic gradient boosting was achieved if the max features hyperparameter or subsampling was a fraction of 1. The hyperparameter learning rate, max depth, max features, and minimum samples and their ranges of testable values were chosen in accordance with recommendations on GBR optimization in scikit-learn [45]. Hyperparameter subsampling was added because Elith et al. [44] showed that stochastic GBR can increase the predictive accuracy. The best hyperparameters were found using cross-validation with prior shuffling of the dataset, five folds on the NV dataset, and always including JV in the calibration dataset. Lastly, the total amount of trees constructed was set at 1000 as recommended by Elith et al. [44]. Manual tuning of the learning rate was performed after the hyperparameter search. The hyperparameters were also set to the GBR models that predicted the lower and upper quantiles.

**Table 3.** Chosen hyperparameter settings after optimization via cross-validation. Note that "Max features" and "Subsampling" are float numbers denoted as a fraction of the total number of features/samples.

| Hyperparameter | Value | Default |
| --- | --- | --- |
| Learning rate | 0.011 | 0.1 |
| Max depth | 6 | 3 |
| Max features | 1.0 | 1.0 |
| Minimum samples | 3 | 1 |
| Subsampling | 0.6 | 1.0 |
| Trees | 1000 | 100 |

*2.9. Cross-Validation and Covariate Importance*

2.9.1. Cross-Validation

Both the PXRF model and the DSM model were validated using a cross-validation approach with five folds on the NV samples. For the DSM model, this was conducted with and without the addition of PXRF-predicted concentrations (i.e., the JV dataset) for calibration; in each iteration, all JV samples were used together with 80% of the NV samples for calibration of the DSM model. The remaining 20% of the NV samples were used to validate the DSM model. This cross-validation design was used to ensure that when the DSM model used the NV and JV dataset together for calibration, the validation was only done for the laboratory-analyzed Cd concentrations. Shuffling was done prior to validation of the DSM model with a different random seed than in the hyperparameter search to obtain folds that were as different as possible. The fold structures of the DSM model validation with and without the PXRF predictions were set to be identical.

2.9.2. Validation Metrics

Two metrics were used to assess the validation performance for the PXRF and DSM model, namely the mean absolute error (MAE) and the Nash–Sutcliffe model efficiency coefficient (E) [46]. An E of 1 would indicate perfect prediction by the model of the soil Cd concentrations in the validation samples. The same metrics were used in the hyperparameter search.

2.9.3. Covariate Importance

In machine learning, the term "feature" is often used for the predictor variables, but the term "covariate" was used in this study in accordance with general DSM terminology. Two different covariate importance assessment methods were used in order to determine the most important covariates for the DSM model. First, since GBR is a tree-based model, it was possible to determine the covariate importance metric mean decrease in impurity (MDI). In short, the MDI indicated how many times a specific covariate was used for a split; the earlier in a tree hierarchy a covariate was used for a split, the more important it was deemed to be [47]. Second, the permutation importance was computed. The permutation importance began by applying the calibrated model on either the calibration data or independent validation data in order to obtain a non-permuted accuracy score (e.g., E), after which one covariate of the input data was permuted (shuffled/corrupted) and the model accuracy was compared against the non-permuted accuracy score—a resulting lower accuracy score implied a more important covariate [47,48]. The shuffling per covariate was conducted a set number of times, and the resulting importance score for each covariate was the mean of those shuffling iterations. In the study, the accuracy score was E, the number of iterations was set at 10, and the model was applied on the calibration data. These two different methods were used on the DSM model that predicted the soil Cd concentrations but not on the DSM models that predicted the lower and upper quantiles (prediction interval).

*2.10. Identifying Areas Suitable for Winter Wheat Production*

Three overlapping subareas in the soil Cd map where the soil Cd concentration was less than or equal to three different limits were delineated. These were used to assess whether the lower predicted concentrations in the soil equaled the lower concentrations in winter wheat grain. The limits set were the 30th, 40th, and 50th percentile of the laboratory-analyzed soil Cd concentration from the NV dataset; i.e., 0.196 mg kg$^{-1}$, 0.215 mg kg$^{-1}$, and 0.240 mg kg$^{-1}$, respectively. The 10th and 20th percentiles were not chosen because very few grain samples were available from areas with soil Cd concentrations below these limits in the produced map. Data on Cd concentrations in winter wheat grain samples within and outside these subareas were compiled and presented in a boxplot. Finally, the approximate proportion of arable land used each year to grow winter wheat within the subareas was calculated using statistics on winter wheat cropping in 2020 (supplied by the Swedish Board of Agriculture, Jönköping, Sweden).

**3. Results**

*3.1. PXRF Modeling and Expanding the DSM Calibration Dataset*

Figure 2 shows the five-fold cross-validation of the PXRF model. Predictions at lower concentrations (0–1 mg kg$^{-1}$) showed no clear bias, whereas predictions at higher concentrations (>1 mg kg$^{-1}$) tended to underestimate the actual soil Cd concentration. The E for the whole cross-validation (0–5.0 mg kg$^{-1}$) was 0.82 with an associated MAE of 0.08 mg kg$^{-1}$. Within the subrange of 0–0.5 mg kg$^{-1}$, the MAE was 0.05 mg kg$^{-1}$. One negative value was predicted for the JV dataset and was set to 0.04 mg kg$^{-1}$, the lowest predicted positive Cd concentration value. Descriptive statistics on the laboratory-analyzed Cd concentrations in the calibration dataset (NV) and the predicted Cd in the JV dataset showed similar values (Table 4). In particular, the percentiles and median were rather similar for the two datasets, while the mean differed more due to skewing caused by higher numbers of observations with high concentrations (Table 4).

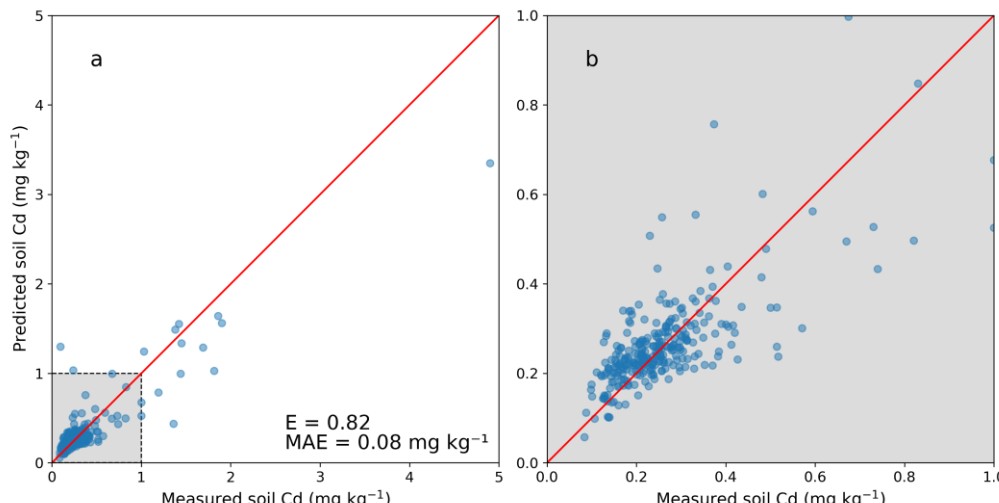

**Figure 2.** (**a**) Cross-validation of the model for predicting cadmium (Cd) concentration in soil from portable X-ray fluorescence (PXRF) measurements and (**b**) enlarged image of the range of 0–1 mg kg$^{-1}$. Five folds were used on 304 samples with laboratory-analyzed Cd concentrations. The Nash–Sutcliffe model efficiency coefficient is displayed (E) as well as the mean absolute error (MAE).

**Table 4.** Descriptive statistics on laboratory-analyzed cadmium (Cd) concentrations in the calibration NV dataset and those predicted by the portable X-ray fluorescence (PXRF) model on the JV dataset (mg kg$^{-1}$).

|  | NV Dataset (Measured Cd) | JV Dataset (Predicted Cd) |
|---|---|---|
| Number of samples | 304 | 2097 |
| Min | 0.08 | 0.04 |
| 25th percentile | 0.18 | 0.21 |
| Median | 0.24 | 0.25 |
| Mean | 0.33 | 0.27 |
| 75th percentile | 0.30 | 0.30 |
| Max | 4.9 | 2.1 |

*3.2. Digital Soil Mapping of Cd Concentration*

Figure 3 shows the cross-validation results of the DSM model with five folds. Compared with predictions using PXRF measurements (Figure 2), the DSM model was less accurate. A positive bias was detected below 0.4 mg kg$^{-1}$ with an associated spread of predicted concentrations. Above the range of 0–1.0 mg kg$^{-1}$, the predictions displayed a negative bias similar to that seen for the PXRF model. The DSM model had an overall E of 0.69 with an associated MAE of 0.11 mg kg$^{-1}$, while in the lower range (0–0.5 mg kg$^{-1}$) the MAE was 0.07 mg kg$^{-1}$. Only using the NV dataset resulted in a less accurate model with an E of 0.58 and an MAE of 0.13 kg kg$^{-1}$. Utilizing the predicted Cd concentrations together with the laboratory-analyzed concentrations thus increased the performance of the DSM model.

Figure 4 shows the most important covariates in the DSM model according to the two different feature importance methods of MDI (Figure 4a) and permutation importance (Figure 4b). Both methods indicated that the airborne gamma-ray measurements (U, Th, and K) were important for the model; the results showed that U was the most important covariate for predicting the soil Cd concentrations. The biogeochemical data also ranked high according to both methods. Elevation and its derivatives were shown to be moderately important; elevation and the different variants of TPI were the most important among these. The soil texture class covariates were rated the least important and showed sandy soils ranking the highest and clay the lowest of the soil texture classes (Figure 4).

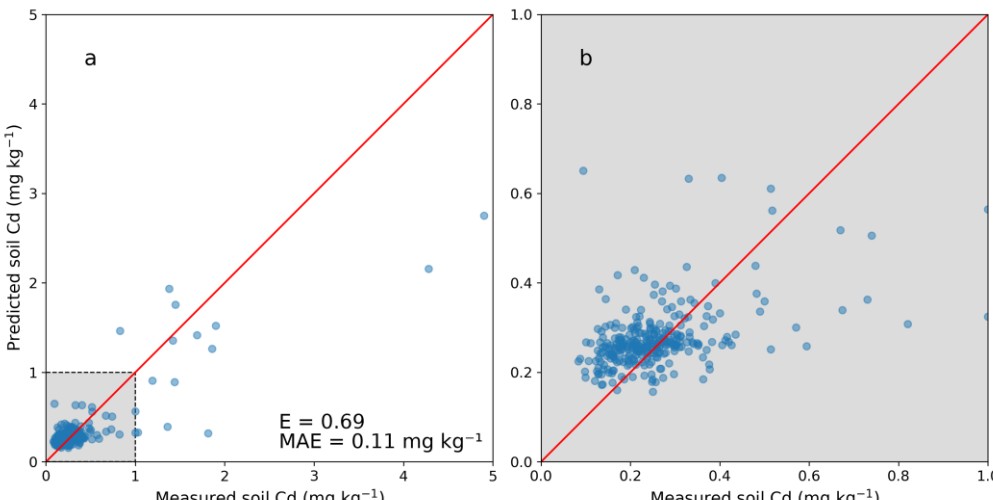

**Figure 3.** (**a**) Cross-validation of the digital soil mapping (DSM) model for predicting cadmium (Cd) concentrations from spatial covariates and (**b**) enlarged image of the range of 0–1 mg kg$^{-1}$. Five folds were used on 304 samples with laboratory-analyzed Cd concentrations. The Nash–Sutcliffe model efficiency coefficient is displayed (E) as well as the mean absolute error (MAE).

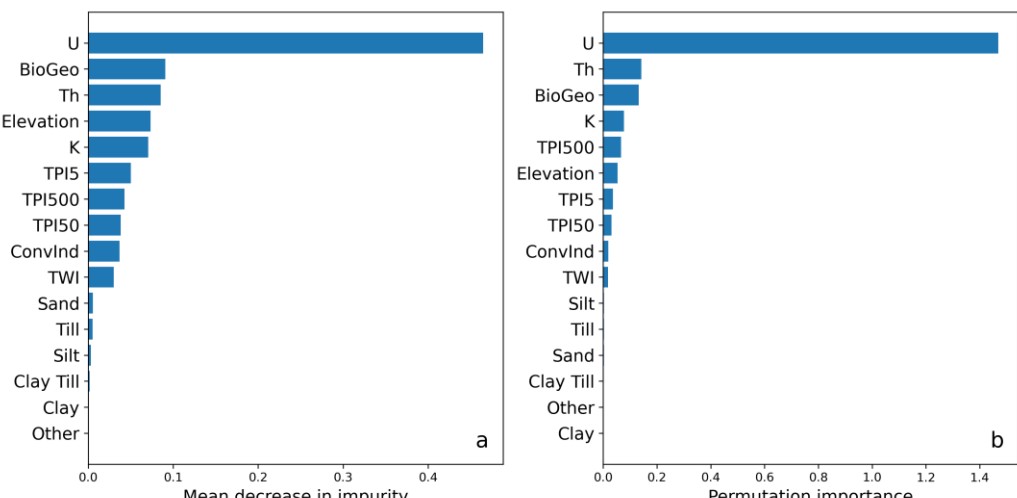

**Figure 4.** (**a**) Covariate importance based on mean decrease in impurity and (**b**) permutation importance of the digital soil mapping (DSM) model. The covariates (see Table 2) were taken from airborne gamma-ray data (U, Th, and K), a digital elevation model (TPI5, TPI50, and TPI500 = topographic position index with 5, 50, and 500 ha radius, respectively; ConvInd = convergence index; TWI = topographic wetness index), biogeochemical data (BioGeo = cokriged with zinc), and quaternary soil deposit classes (sand, till, silt, till clay, clay, and other). Higher values indicate higher covariate importance.

The final soil map of Cd concentrations and the 90% prediction interval revealed that areas with low predicted Cd concentrations and narrow intervals were mainly located in the northwest, northeast, and center of Skåne (Figure 5). Areas with high predicted Cd concentrations were mainly in the southeast and in topographical depressions such as some river valleys. Interestingly, isolated areas with low predicted Cd concentrations were indicated across Skåne. In general, areas with high predicted Cd concentrations had a wide prediction interval and thus higher uncertainty, while areas with low predicted Cd concentrations often but not always exhibited a narrow prediction interval. An artifact in the form of a horizontal line was visible in the southeast that was mainly due to the U covariate layer exhibiting a contrasting line derived from different measurement dates and equipment.

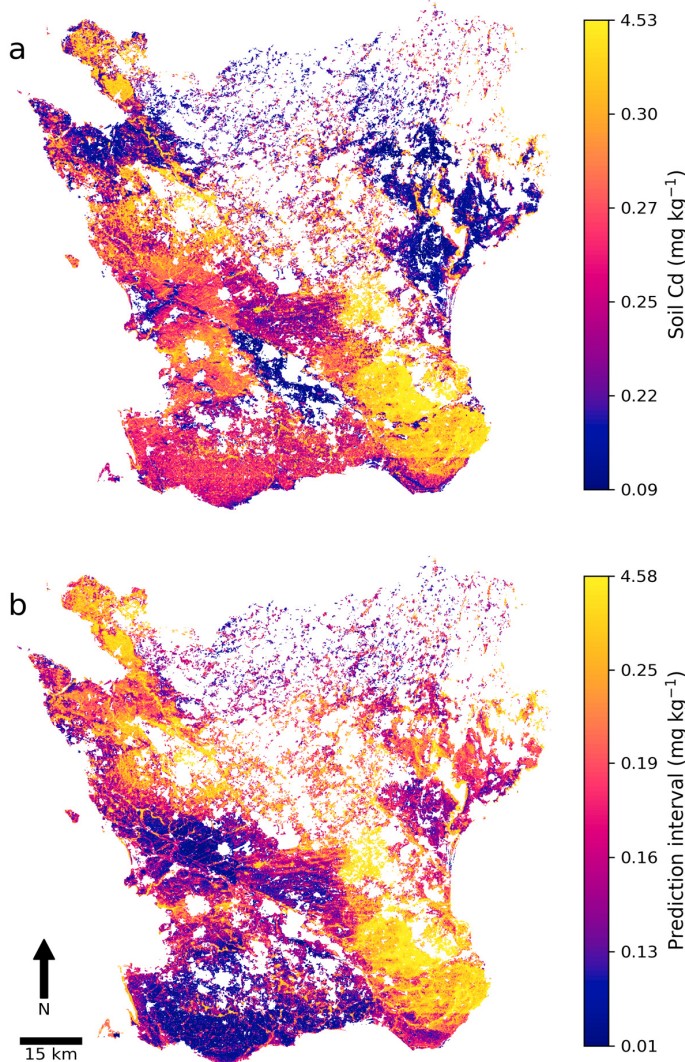

**Figure 5.** (**a**) Predicted cadmium (Cd) concentrations (mg kg$^{-1}$) in agricultural topsoil in Skåne and (**b**) the 90% prediction interval width.

Descriptive statistics on the two digital soil maps revealed that the predicted concentrations and prediction intervals were positively skewed (Table 5). In both maps, the minimum and maximum were well above or below the respective 25th and 75th percentile, meaning that areas with very low predicted Cd concentrations or a narrow 90% prediction interval were uncommon. The same was true for areas with high predicted Cd concentrations and a wide 90% prediction interval. Compared against the laboratory-analyzed Cd concentrations of the NV dataset (Table 4), the digital map of soil Cd concentrations had a fairly similar 75th percentile and median.

**Table 5.** Descriptive statistics on the digital soil map (DSM) of cadmium (Cd) concentrations and the 90% prediction interval map (mg kg$^{-1}$).

|  | Min | 25th Percentile | Median | Mean | 75th Percentile | Max |
|---|---|---|---|---|---|---|
| Prediction | 0.09 | 0.23 | 0.26 | 0.28 | 0.29 | 4.53 |
| Prediction interval | 0.01 | 0.14 | 0.17 | 0.21 | 0.22 | 4.58 |

*3.3. Digital Soil Map versus Grain Concentrations*

Figure 6 shows Skåne subdivided into areas with predicted Cd concentrations below and above the three different limits and box-and-whisker plots of the Cd concentration in winter wheat grain samples collected within the three delineated areas. The majority of grain samples were from areas with Cd concentrations higher than the limit set. Many grain samples were from the west, southwest, and southeast of Skåne (Figure 1), where soil Cd concentrations are generally above the set limits (Figure 6). The box-and-whisker plots showed that areas with a low predicted soil Cd concentration also had a lower median Cd concentration in grain samples. Grain samples with a low Cd concentration were also present in areas with a high predicted soil Cd, but high grain Cd concentrations were less common in areas with a low predicted soil Cd.

Table 6 shows the approximate area of winter wheat production in Skåne in 2020 below each soil Cd limit shown in Figure 6. In that year, which served as an example, the total area of winter wheat production in Skåne was 101,383 ha. The area with winter wheat more than doubled when the limit was increased from 0.215 mg kg$^{-1}$ to 0.240 mg kg$^{-1}$. Only 4.4% of the total winter wheat was produced within the subarea below the lowest limit (Table 6).

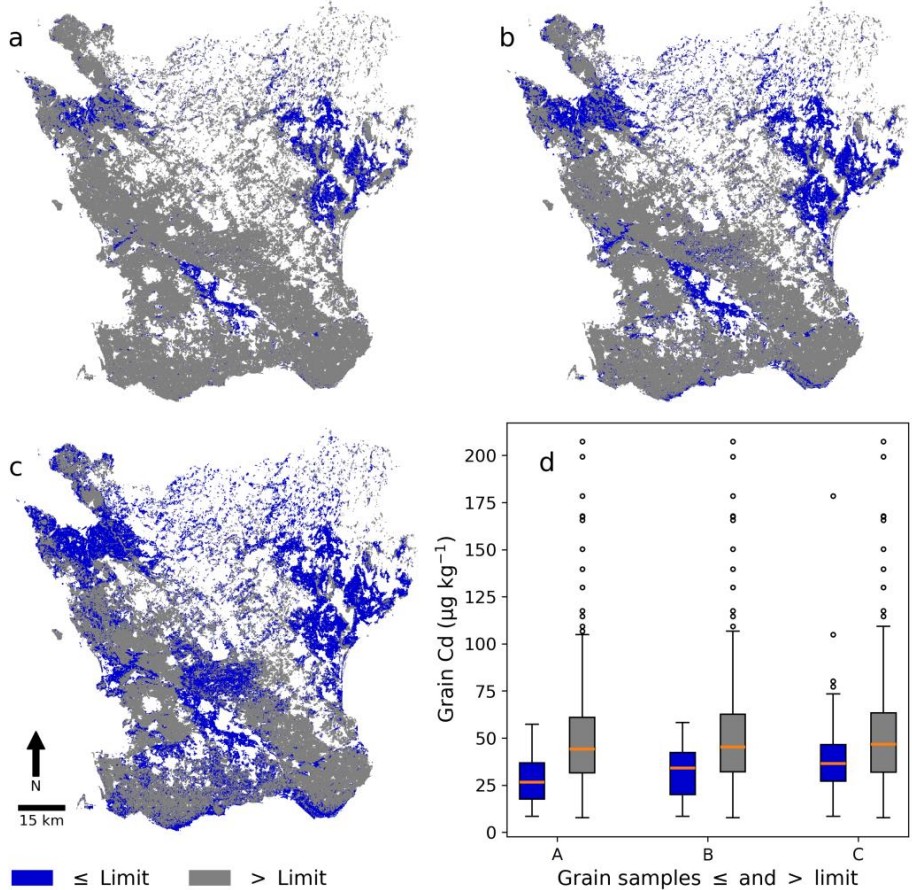

**Figure 6.** (**a–c**) Areas in the digital soil map (DSM) with cadmium (Cd) concentrations below (blue) and above (grey) three different limits and (**d**) box-and-whisker plots of Cd concentration in winter wheat grain in these areas. The limit in (**a**), (**b**), and (**c**) was set at 0.196 mg kg$^{-1}$, 0.215 mg kg$^{-1}$, and 0.240 mg kg$^{-1}$, respectively, which corresponded to the 30th, 40th, and 50th percentile of laboratory-analyzed soil Cd concentrations in the NV dataset. The numbers of winter wheat grain samples in blue boxes and whisker plots A, B, and C are 17, 34, and 75 from a total of 307. Dots represent outliers, the orange line the median, the box the interquartile range (IQR), and the whiskers the highest/lowest data point within the IQR × 1.5.

**Table 6.** Area of winter wheat production in 2020 (Swedish Board of Agriculture, Jönköping, Sweden) within areas of the digital soil map (DSM) with cadmium (Cd) concentrations below the limits corresponding to the 30th, 40th, and 50th percentile of laboratory-analyzed soil Cd concentrations in the NV dataset.

| Limit Concentration in Soil (mg kg$^{-1}$) | Area of Wheat Production in 2020 (ha) |
|---|---|
| 0.196 | 4486 |
| 0.215 | 9373 |
| 0.240 | 20,299 |

## 4. Discussion

### 4.1. PXRF Modeling

Models for prediction of Cd from the PXRF measurements were deemed sufficiently accurate to extend the calibration dataset for the DSM. The performance of the PXRF model was similar to that of the best-performing PXRF model previously presented by Adler et al. [16] in terms of the MAE ($\approx$0.05 mg kg$^{-1}$). However, the E was marginally higher in the present study. This could perhaps be attributed to the PXRF model being a combination of the RF and MARS algorithms and to it being a regional model rather than the national model used by Adler et al. [16]. The accuracy achieved by the DSM model indicated that Cd predicted from the PXRF measurements together with the laboratory-analyzed Cd concentrations could be used as calibration data. Using the PXRF-predicted and laboratory-analyzed concentrations together made the DSM model more accurate than only using the laboratory-analyzed concentrations (E = 0.69 and 0.58, respectively). The increase in performance was probably due to the geographically denser calibration dataset for the DSM model compared to only using the laboratory-analyzed concentrations. However, the predictions of the PXRF model were not completely in agreement with the laboratory-measured values, which was probably due to the inherent uncertainties in the laboratory analysis and the accuracy of the PXRF model. Future research should assess the latter by determining the optimal machine learning algorithm and covariates (i.e., the most suitable elements) and perhaps using hyperparameter optimization to reduce this disparity further.

### 4.2. Digital Soil Mapping

Areas with low predicted Cd concentrations in the northwest and northeast of the study area (Figure 5a) were generally dominated by postglacial sediments and those in the center by glaciofluvial deposits. The areas with the highest predicted Cd concentrations (in southeastern Skåne) were previously identified as high-risk areas by Söderström and Eriksson [15]. As shown by the cross-validation results (Figure 3), the DSM model exhibited a lack in performance in terms of the MAE and E (0.11 mg kg$^{-1}$ and 0.69, respectively). In regression kriging of Cd in soil with similar covariates for an area in the Hunan province in China, Cao et al. [49] achieved a similar accuracy in terms of the E and MAE using leave-one-out cross-validation. Hence, accurately mapping soil Cd seems to be a difficult endeavor.

Low grain Cd concentrations were associated to a higher degree with areas with low predicted low soil Cd concentrations in this study than to areas with high predicted soil Cd concentrations (Figure 6). However, there were still areas with predicted high soil Cd concentrations that showed low grain Cd concentrations (Figure 6). Areas delineated by the lower limits of Cd concentration in soil (0.196 and 0.215 mg kg$^{-1}$) had few samples with grain Cd concentrations above 50 µg kg$^{-1}$. These results showed that a large part of the soils in areas with low Cd concentration in the soil Cd map could be safe to use for the production of winter wheat if the specific aim is low grain Cd concentrations. The soil Cd map could be especially interesting when used together with the 90% prediction interval map; e.g., for a delineated area with low soil Cd concentration, the 90% prediction interval map could be used to determine whether this predicted concentration was reliable. The prediction interval map could thus be used for more precise identification of areas suitable

for production of winter wheat with low Cd concentration by selecting only those areas with a low enough uncertainty; i.e., a narrow prediction interval. However, conveying uncertainty is difficult and what is deemed accurate (or not) is dependent on the end user or application [50]. The accuracy of the DSM model and the uncertainty indicated that verification of the soil Cd map is advisable, especially if the predicted concentration is close to a specific limit or if absolute certainty is needed. The soil Cd map should thus be regarded as an explorative or supportive tool within a decision-support framework. An interesting future task could be to expand and use the prediction interval map in a meaningful way together with the soil Cd map. If the uncertainty were to be used, some kind of validation of it would be needed [51] by using, e.g., the percentage coverage of cross-validation predictions within the prediction interval (prediction interval coverage probability (PICP)) [52].

### 4.3. Covariate Importance in Digital Soil Mapping

The covariate importance assessments revealed the significance of airborne gamma-ray data—specifically U—for the DSM model, while soil texture classes were of low significance. This raised questions regarding the information contained in each covariate. Airborne gamma-ray measurements of agricultural soil mainly contain information about the type of soil parent material and soil texture in the upper 0.3 m of the soil profile [53]. Hence, the soil classes probably contained redundant and less accurate information in this study due to its focus on the soil texture at a 0.5 m depth. This redundancy was also evident to a degree with the DEM derivatives because elevation and the different TPIs were ranked higher than ConvInd and TWI (Figure 4). A handful of derivatives were used in the study, but many more can be created via different methods of calculation and execution. For instance, TWI can be calculated using different methods with similar but not identical results [54]. Moreover, airborne gamma-ray measurements can be affected by soil moisture, which indicates a need to include moisture-related DEM derivatives when airborne gamma-ray data are used in order to potentially correct for this [53].

Biogeochemical data proved to be an important covariate. Much more could be done with this covariate in the future because it can be processed in a vast number of ways using various methodologies. However, using aquatic plants to obtain a rough delineation of Cd in the study area seemed to be a useful covariate for mapping the soil Cd concentrations.

In future studies, it would be interesting to attempt to include covariates related to human activities such as crop rotation, amounts of phosphorus applied, and aerial deposition of Cd. However, a difficulty may be that such factors have affected soils over several decades, and it is often difficult to find reliable data on this for longer time periods. Hence, in the present study, it was not possible to include covariates that represented human activities because no such dataset was available.

### 4.4. Potential Use in Identifying Suitable Areas for Production of Winter Wheat with Low Cd Grain Concentration

The results showed that winter wheat fields with low ($<0.2$ mg kg$^{-1}$) soil Cd concentrations are relatively uncommon in Skåne (Table 6). Attention should perhaps be focused on maintaining as favorable a soil Cd concentration as possible in these fields. Areas in northwest, northeast, and central Skåne could be the most favorable for production of winter wheat with low Cd concentrations (Figures 5 and 6). Hence, the soil Cd map could be used as a tool for identifying areas generally suitable for sourcing or producing winter wheat grain with low Cd concentrations; e.g., for use in baby food manufacturing. Suitable areas for other interesting cereals for baby food, such as oats or durum wheat (*Triticum durum* Desf.), could potentially be assessed in a similar way. However, further testing is needed to determine whether the soil Cd map is sufficiently accurate to delineate within-field boundaries, thereby allowing a certain part of a field with low soil Cd concentrations to be allocated to winter wheat production.



## 5. Conclusions

In this study, PXRF measurements were used to predict (E = 0.82) the soil Cd concentrations in samples with an unknown Cd concentration, which made it possible to enlarge the calibration dataset from 304 to 2401 samples for the subsequent DSM model. The results showed that this calibration dataset made the DSM model more accurate than when only using soil samples with laboratory-analyzed concentrations. The DSM model (E = 0.69) produced a soil Cd map of Skåne that could be used to identify areas with low soil Cd concentrations. The available data confirmed that winter wheat grain Cd concentrations were generally low in these areas. A 90% prediction interval map was also created that could be used to assess the reliability of predictions. A covariate importance analysis revealed that airborne gamma-ray data (specifically U), cokriged biogeochemical data, and certain elevation derivatives were the most important covariates in the DSM model. The soil Cd map indicated that areas in northwest, northeast, and central Skåne were the most suitable for producing winter wheat with low soil Cd concentrations.

**Author Contributions:** Conceptualization, K.A., K.P. and M.S.; methodology, K.A.; software, K.A.; validation, K.A.; formal analysis, K.A.; investigation, K.A.; resources, K.A., K.P., M.S. and J.E.; data curation, K.A., K.P., M.S. and J.E.; writing—original draft preparation, K.A.; writing—review and editing, K.A., K.P., M.S., J.E. and C.-G.P.; visualization, K.A.; supervision, K.P., M.S., J.E. and C.-G.P.; project administration, K.A. and K.P.; funding acquisition, K.P. and M.S. All authors have read and agreed to the published version of the manuscript.

**Funding:** This research was funded by the Västra Götaland Region and the Swedish University of Agricultural Sciences (grant number: 2018-00141) and Formas (grant number: 2019-02280).

**Data Availability Statement:** Digital elevation data are available from the Swedish National Land Survey (Lantmäteriet) and gamma ray and biogeochemical data from the Geological Survey of Sweden (SGU). The datasets of the Swedish Board of Agriculture (JV) and Swedish Environmental Protection Agency (NV) administered by the Swedish University of Agricultural Science are not publicly available.

**Acknowledgments:** We would like to thank the Geological Survey of Sweden (SGU), the Swedish Board of Agriculture (JV), the Swedish Environmental Protection Agency (NV), and Lantmäteriet for supplying data. We would also like to thank Omran Alshihabi, Swedish University of Agricultural Sciences, for the PXRF measurements of the soil samples.

**Conflicts of Interest:** The authors declare no conflict of interest.

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
