# Peer review of "Digital Soil Mapping of Cadmium: Identifying Arable Land for Producing Winter Wheat with Low Concentrations of Cadmium"

_agronomy, doi:10.3390/agronomy13020317_

Round 1

Reviewer 1 Report

The authors write about their experience in predicting Cadmium (Cd) concentrations in the topsoil of agricultural areas in southern Sweden. They first predict topsoil Cd concentrations for a PXRF-only measured dataset using model averaging with RF and MARS methods. Then, they utilize the predicted Cd concentrations resulting from the first model to expand a calibration dataset to map topsoil Cd concentrations over the agricultural areas of Skåne using gradient boosting and different environmental covariates (following the DSM approach).

The manuscript is well-written and the topic fits with the scope of the journal. However, relatively major revision is required for the manuscript to be considered ready for publication. The main point to revise is the way cross validation is used for one of the DSM models. The author should use cross validation in a proper way, not artificially keeping a large part of the dataset for calibration only (a clear bias). I strongly recommend comparing models based on cross validation with on the one hand the predictions using PXRF measurements (i.e., JV dataset only), and on the other hand the wet chemistry and PXRF measurements (i.e., NV dataset only). Another important point developed below is that the abstract needs rewriting. These two main points are further developed below, and specific and detailed comments can also be found.

Abstract:

-        A first recommendation would be to start the abstract with a few sentences giving a short background/context to your study, why it was carried out and why it matters.

-        Line 9: regression rather than regressor

-        Line 11: soil rather than Soil

-        Lines 11 and 12:  you could simplify the sentence ”the soil samples were analyzed … device and  304 samples …”

-        Would you clarify your modelling process? First, you model Cd concentrations using RF/MARS with PXRF data (304 samples with wet chemistry for training and validation, prediction on the remaining 2097 samples) and then you predict the same target in the geographical space using gradient boosting with all the soil samples (for training and validation), and 307 winter wheat grain samples as an independant validation dataset, is it right?

As a general comment, the abstract is the first section of a paper that possible readers go through, so it is really important to get all the key points of your study in it, and as clearly as possible. You want to catch the interest of your readers straight.

Introduction:

-        What are the possible sources of Cadmium in the soil? Could you add here a few sentences and references? There is one sentence and a reference in the study area description in the next section, but it is too late and too little (what soil parent material? How does it end up in the soil? As a geologist I have my idea, but what about readers without this background knowledge).

-        Your specific objectives are clear and should be already introduced in the abstract.

-        In the introduction, you only mention machine learning, I would recommend you to be more specific (what methods were used in Adler et al (16) and in the present study, this is important to state it here already)

-        Line 77: ”a detailed map of soil Cd”

-        Lines 77 and 78: you can simplify ”with a 50-m spatial resolution”

Mat & Met:

-        Line 86: ”the study area is”

-        Can you add the extent of your study area?

-        Figure 1: Clarify NV (Naturvårdsverket) and JV (Jordbruksverket) in the caption, and ”for which laboratory-measured cadmiun data were used to…”

-        Line 123: ”(0-20 cm depth)”

-        When were the samples collected? Just to have an idea in comparison with the grain samples from 1992 and 2001-2017.

-        Line 132: The 304 NV samples were used for calibration only of the PXRF model? How did you validate your model then?

-        Line 180: ”in a 50-m resolution grid”

-        Table 2: in the caption, when you write separate covariate, do you mean dummy/indicator/one-hot-encoded covariate?

-        Line 193: ” A 2-m resolution DEM”

-        Line 194: ”to 10-m resolution”

-        Line 196: you should avoid using i.e. outside of brackets

-        Lines 197-199: you can clarify how you generated the two derivatives (SAGA?)

-        Lines 227-229: This sentence belongs to the introduction (background information with a similar and pre-existing study). This information concerning the models used to model Cd concentrations from PXRF data should be given in the introduction.

-        Line 232: Can you rewrite to avoid the redundancy of ”distinctly different”?

-        Lines 233-234: what kind of testing/optimization? Grid search? Bayesian optimization? What are the default settings?

-        Lines 235-236: Can you clarify how you evaluated the performance of your models if the whole NV dataset was used for calibration only? Later you state that you used cross-validation on the NV dataset which contradicts the sentence here.

-        Line 251: ”the hyperparameters…were…”

-        Lines 274-277: what you describe here is not cross validation. In CV, samples used for calibration are used at least once for validation. You are only cross validating on the NV dataset while still using JV dataset for calibration, and this can only result in biased models, especially as the JV dataset is much larger than the NV dataset. I would recommend to use the JV dataset for calibration and the NV for validation of a model, or better to use the JV dataset alone for cross-validation of another model and then use the NV as an independant test dataset.

-        Line 283: this performance metrics is not so often used, I would suggest clarifying its calculation with the corresponding equation.

Results

-        Line 348: artificially increased with their present use I am afraid.

-        Table 4: seeing how skewed the measured Cd values are, have you considered log transforming the data before modelling (to get closer to a normal distribution)? This is a simple step, one only has to back transform both observations and predictions before calculating performance metrics.

-        The comparison of models does not make sense if you use a dataset as calibration only, without putting part of it to the test. This comparison is unfair: in the models using both JV and 80% of NV, you have a larger dataset and you don’t test the accuracy of your model on any JV subset.

Considering my earlier comment on the matter, I would recommend using five fold cross-validation on the one hand with the JV dataset only and on the other hand with the NV dataset only, then the comparison of performances would make more sense. You could also check the prediction of the first model on NV as an independant test dataset.

Discussion

-        Lines 432-434: The statement here is for now based on a poor comparison between DSM models. The incorrect use of cross validation in the model combining JV and NV datasets does not allow to make such a conclusion. I guess you would get better results with a model using only JV dataset and most probably even better with a combination of JV and NV datasets, in comparison to NV dataset alone. But cross validation has to be used in a proper way.

Author Response

Dear reviewer, thank you for your input! Please look at the attached file with our responses.

Reviewer 2 Report

The study developed a gradient boosting regressor model to map soil Cd distribution in south Sweden. To improve the DSM accuracy, a PXRF model was designed to determine soil Cd content in 2097 soil samples, this is a good method to enlarge sample size with relatively less workforce and expenditure.

But there are some concerns:

1)      The gradient boosting regressor can be compared with traditional models, such as kriging. The result may be better when using kriging model with more than 2000 sampling points. You can compare boosting regressor with other models in supplementary materials.

2)      The human activities can be considered as covariates, such as fertilizers and pesticides usage in agricultural land. You can retrain the model with new features or discuss the human activities in discussion part if this type of features is hard to acquire.

3)      In Section 3.3, a statistical test is preferred to confirm if there is significant difference between two groups of wheat grain.

4)      Cadmium concentrations of some soil samples are tested with the Niton XL3t GOLDD+ PXRF device ex situ. I think when you test the reference sample National Institute of Standards and Technology (NIST) 2709a, the reference sample must be dried and milled. When test the soil sample ex situ, how much does the scattering influence caused by the soil moisture on X-ray fluorescent intensity of Cd in the soil sample?

Author Response

Dear reviewer! Thank you for your input! Please see the attached file with our responses!
